# Learning While Deploying: Fleet-Scale Reinforcement Learning for Generalist Robot Policies

*Abstract*—Generalist robot policies increasingly benefit from large-scale pretraining, but offline data alone is insufficient for robust real-world deployment. Deployed robots encounter distribution shifts, long-tail failures, task variations, and human correction opportunities that fixed demonstration datasets cannot fully capture. We present *Learning While Deploying* (LWD), a fleet-scale offline-to-online reinforcement learning framework for continual post-training of generalist Vision-Language-Action (VLA) policies. Starting from a pretrained VLA policy, LWD closes the loop between deployment, shared physical experience, policy improvement, and redeployment by using autonomous rollouts and human interventions collected across a robot fleet. To stabilize learning from heterogeneous, sparse-reward fleet data, LWD combines *Distributional Implicit Value Learning* (DIVL) for robust value estimation with Q-learning via *Adjoint Matching* (QAM) for policy extraction in flow-based VLA action generators. We validate LWD on a fleet of 16 dual-arm robots across eight real-world manipulation tasks, including semantic grocery restocking and 3–5 minute long-horizon tasks. A single generalist policy improves as fleet experience accumulates, reaching an average success rate of 95%, with the largest gains on long-horizon tasks.

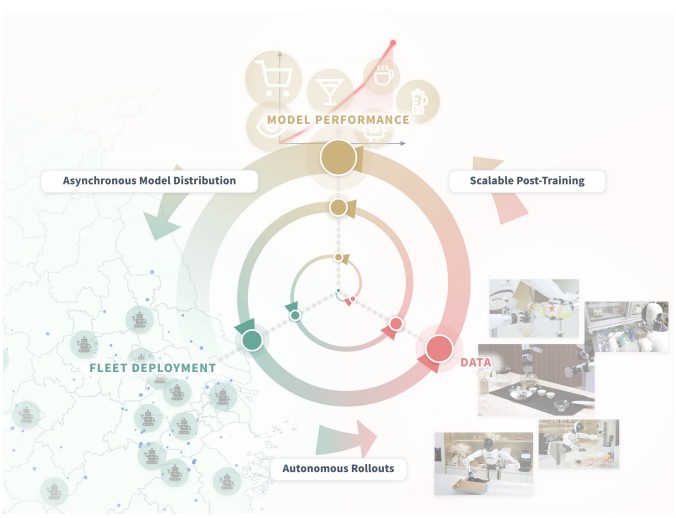

Fig. 1. **Learning While Deploying.** A fleet autonomously collects deployment data, mixes it with offline replay, updates the shared VLA policy, and redeploys the improved model.

## I. INTRODUCTION

Deploying general-purpose robots in the real world requires *high-performance generalist* policies: policies that can reliably complete a broad range of tasks across diverse objects, environments, user instructions, and operating conditions. Recent Vision-Language-Action (VLA) policies [8, 62, 52, 20, 5, 6] provide a strong foundation by acquiring broad competence from large offline robot datasets. However, offline pretraining alone does not make a policy deployment-ready. Real-world deployment is not a fixed test distribution: as robots are used across more homes, stores, workspaces, and users, they encounter new tasks, object instances, configurations, preferences, and rare failure modes beyond the coverage of pretraining data. Obtaining high performance therefore requires policies that continue to improve from deployment experience, so that adaptation scales with the data generated by use.

This perspective recasts deployment from an endpoint of training into a source of continual policy improvement. Realizing this form of continual improvement requires deployment experience that is both broad and continuously updated. For a generalist robot policy, the most valuable deployment experience is naturally collected at fleet scale. Any individual robot samples only a small portion of the deployed distribution, whereas a fleet spans diverse tasks, environments, objects, and user instructions, producing heterogeneous experience that includes successes, failures, recoveries, partial progress, rare

edge cases, and occasional human interventions. Aggregating this physical experience through a shared policy creates a closed-loop data flywheel: deployed robots generate experience on the target deployment distribution, the shared policy improves from the aggregated data, and the improved policy is redeployed to collect broader and more informative experience.

We refer to this setting as *Learning While Deploying* (LWD): continual policy improvement driven by the accumulated real-world autonomous experience of a deployed robot fleet. Turning this data flywheel into a learning algorithm, however, requires a training objective that can improve from the outcomes of autonomous interaction, rather than treating deployment data for pure imitation signal. Interactive imitation-learning methods [19] can incorporate expert demonstrations, corrections, and interventions during deployment, but they treat deployment primarily as a source of action labels for supervised learning. As a result, they use only part of the available experience and lack a principled mechanism for leveraging autonomous trials that contain successes, failures, recoveries, partial progress, and task rewards. Reinforcement Learning (RL) in principle provides such a mechanism by optimizing policy behavior from task outcomes and policy experience [54, 14, 32, 15]. Yet existing RL approaches for robotics are often limited to small-scale, short-horizon, or task-specific settings, and frequently specialize

a pretrained generalist policy to a narrow task [26, 31, 10]. A scalable method for post-training end-to-end VLA policies from fleet deployment experience while preserving their generality remains an open problem.

Addressing this gap requires an RL algorithm for LWD that is compatible with pretrained VLA policies, can learn from large offline and off-policy datasets, and can adapt rapidly as new deployment data streams in. These requirements stress both components of an RL method. Value learning must produce reliable estimates from heterogeneous off-policy data with sparse rewards and rare high-return trajectories. Policy extraction must turn the learned values into better actions from a large generative VLA policy without destabilizing the model.

Prior work addresses these requirements only in part. Amin et al. [2] combines offline value learning with iterative offline RL, but the procedure is slow and does not directly use action gradients from the learned value function. Luo et al. [39, 40] show that online RL can learn challenging robotic manipulation tasks within a short period through real-world interaction, but train task-specific policies from scratch rather than improve a pretrained generalist policy. On-policy VLA finetuning methods [38, 50, 9, 35, 60] update pretrained policies directly from online rollouts, but are not designed to efficiently reuse large offline or off-policy deployment buffers. They also do not learn an explicit action-value critic, and therefore cannot use action-space critic gradients to guide policy improvement. Together, these limitations motivate an offline-to-online RL approach that can reuse heterogeneous deployment data while stably improving a pretrained generative VLA policy.

We present Fleet-Scale Offline-to-Online RL, an offline-to-online framework for post-training end-to-end VLA policies in a large-scale real-world deployment system. The framework couples two pieces: distributional value learning from offline and autonomous deployment experience, and stable policy extraction that transfers value improvement into a flow-based VLA policy.

For value learning, we introduce Distributional Implicit Value Learning (DIVL). DIVL builds on the value-learning component of Implicit Q-Learning [21], but replaces scalar expectile value regression with a distributional value model. This choice is important in the setting of fleet deployment since robots collect diverse data asynchronously under a variety of conditions. As a result, the return associated with the same state-action pair can be multi-modal and heavy-tailed. A scalar critic may collapse these outcomes into an average value and obscure rare but reproducible successes, whereas a distributional critic can preserve these high-return modes. DIVL therefore learns multi-step return distributions while retaining the in-support policy improvement property of implicit value learning. This yields a stable learning signal from large off-policy deployment buffers without requiring the policy to query out-of-distribution actions.

For policy extraction, we adopt Q-learning with Adjoint Matching (QAM) [11, 30]. The critic provides useful action gradients, but backpropagating them through the full multi-step denoising process of a flow policy is unstable and expensive.

QAM converts the critic gradient at the denoised action into step-wise supervision for the flow model. This gives a stable way to update the VLA policy from the learned value function while preserving the expressivity of generative action modeling.

The full system has two stages: offline pretraining on a mixture of data from diverse sources, followed by rapid online finetuning with deployment data. Both stages optimize the same RL objective, which mitigates a common offline-to-online mismatch: offline critics can become overly conservative and poorly calibrated for subsequent online finetuning, while online improvement depends on extrapolating values to newly visited actions [44]. We instantiate it on a fleet of 16 dual-arm robots across eight manipulation tasks. These include long-horizon precision tasks, such as brewing Gongfu tea, making cocktails, and making fruit juice, which typically require 3–5 minute executions, as well as shorter-horizon tasks that require semantic generalization, such as restocking diverse items in grocery stores. A single generalist policy trained with LWD improves as online fleet experience accumulates. It substantially improves over the pretrained model, reaches an average success rate of $0.95$ across all tasks, and outperforms relevant baselines by large margins. The performance gap is especially pronounced on long-horizon tasks, where RL can propagate rewards through multi-step dynamic programming and stitch together value estimates across partial progress, while imitation-learning methods suffer more severely from compounding errors. This LWD procedure typically requires only a few hours of real-world interaction.

Our main contribution is a fleet-scale offline-to-online RL system for post-training generalist robot policies in real-world deployment. Algorithmically, LWD combines distributional implicit value learning with QAM-based policy extraction and uses the same RL objective across offline pretraining and online finetuning. Systemically, it enables a distributed robot fleet to aggregate physical interaction experience and autonomously improve a shared VLA policy. To the best of our knowledge, LWD is among the first real-world RL systems to close this offline-to-online improvement loop for generalist robot policies. More broadly, it provides a concrete step toward deploying general-purpose robots at scale: fleet-scale deployment can itself become a source of training data, creating a data flywheel in which deploying more robots improves the shared policy and, in turn, future deployment.

## II. RELATED WORK

*a) Post-Training of Robot Generalist Policies:* Robot generalist policies, including VLA models, acquire broad capabilities through large-scale pre-training on diverse data [62, 52, 20, 6]. To adapt these policies to deployments, recent work has explored post-training strategies [61, 9, 26, 2, 56, 36]. One direction studies offline post-training, where policies are improved using pre-collected rollouts [61, 2, 55]. $\pi_{0.6}^*$ combines offline value learning with iterative offline RL to improve individual real-world tasks [2]. RLDG uses specialist RL to generate data for policy distillation [55]. However, offline-only methods follow a collect-train-deploy cycle, facing

distribution shifts during deployment [2, 55]. LWD instead updates the policy during deployment, allowing new experience to correct such shifts quickly. Another line of work performs online RL or online post-training, including VLA-RL [38] and RIPT [50], achieving strong gains for specialist policies in simulation [27, 41, 34, 42, 9, 57]. However, these methods mainly rely on on-policy data, making real-world training sample-inefficient and costly [9, 28]. LWD differs from both offline-only and online-only post-training: it uses deployment as a continual learning loop for a single generalist policy, while reusing offline data and off-policy online replay for real-world training.

*b) Offline-to-Online Reinforcement Learning:* Offline-to-online RL connects offline experience with online improvement [46, 30, 23, 25, 1, 3]. One class of methods uses offline data to initialize online finetuning, often by seeding replay buffers or constraining policy updates. Luo et al. [39, 40] use a small number of demonstrations to speed real-world policy learning, while methods such as AWAC reuse offline data during online improvement [43, 49]. These approaches improve sample efficiency, but are typically demonstrated on single skills rather than generalist policies. Another class first performs offline training and then online finetuning, such as ConRFT, RL-100, DSRL and QAM [31, 10, 26, 55, 53, 30]. This setting is closer to LWD, but existing methods usually focus on task-specific policies, use different objectives across offline and online stages, or operate at limited scale. In contrast, LWD post-trains a single generalist policy across multiple tasks in distributed fleet-scale deployment, using a unified RL objective over offline data and off-policy online replay.

*c) Large-Scale Robotic RL Systems:* Large-scale robotic RL systems improve policies by aggregating experience from distributed actors with centralized training, enabling learning beyond isolated task-level data collection [17, 18, 24, 45, 7, 13, 16]. Kalashnikov et al. [17, 18] demonstrate that off-policy RL can be scaled from vision-based grasping to multi-task manipulation through asynchronous robot data collection and centralized Q-function optimization. Bousmalis et al. [7] and Herzog et al. [16] further study learning from large-scale robot experience, but the former relies on behavior cloning while the latter targets task-specific RL for waste sorting. More recently, SOP [45] formalizes a scalable system for online VLA post-training with robot fleets, centralized learning, and asynchronous policy synchronization. Building on this substrate, LWD applies offline-to-online RL to leverage offline data and fleet experience for improving a generalist policy across real-world tasks. This contributes to an RL-driven data flywheel, where large-scale deployment continually supplies experience for policy improvement.

## III. PRELIMINARIES

### A. Problem Setting and Notation

We formulate robot control as a Markov decision process $\mathcal{M} = (\mathcal{S}, \mathcal{A}, \mathcal{T}, r, \gamma)$. Each state $s = (o, \ell) \in \mathcal{S}$ consists of a robot observation $o$ and task-dependent language instruction $\ell$.

We use sparse binary rewards, with $r = 1$ only for successful terminal states and $r = 0$ otherwise.

LWD trains one generalist VLA policy across all tasks. At time $t$, the policy outputs an action chunk

$$\mathbf{a}_t \equiv \mathbf{a}_{t:t+H} = [a_t, a_{t+1}, \ldots, a_{t+H-1}] \sim \pi_\theta(\cdot \mid s_t), \quad (1)$$

which is executed before replanning. The corresponding chunk reward is

$$\mathbf{r}_t \equiv \mathbf{r}_{t:t+H} = \sum_{i=0}^{H-1} \gamma^i r_{t+i}. \quad (2)$$

Replay samples are written as $(s_t, \mathbf{a}_t, \mathbf{r}_t, s_{t+H}) \sim \mathcal{D}$, where $\mathcal{D}$ is induced by $\mathcal{B}_{\text{off}}$ in the offline stage and by mixed replay from $\mathcal{B}_{\text{off}} \cup \mathcal{B}_{\text{on}}$ in the online stage.

### B. Implicit Q-Learning

Implicit Q-Learning (IQL) [21] avoids explicit action maximization by fitting a *scalar* state-value function to a high *expectile* of dataset action-values. Using the chunk notation from above, IQL fits

$$\mathcal{L}_V^{\text{IQL}}(\psi) = \mathbb{E}_{\mathcal{D}}\left[\rho_{\tau,2}(Q_{\bar{\phi}}(s_t, \mathbf{a}_t) - V_\psi^{\text{IQL}}(s_t))\right], \quad (3)$$

where $\rho_{\tau,2}(u) = |\tau - \mathbb{I}(u < 0)|\, u^2$ and $Q_{\bar{\phi}}$ denotes the EMA target network. The critic $Q_\phi$ is trained with the value-based TD loss

$$\mathcal{L}_Q^{\text{IQL}}(\phi) = \mathbb{E}_{\mathcal{D}}\left[\left(Q_\phi(s_t, \mathbf{a}_t) - y_t^{\text{IQL}}\right)^2\right], \quad (4)$$

$$y_t^{\text{IQL}} = \mathbf{r}_t + \gamma^H V_\psi^{\text{IQL}}(s_{t+H}). \quad (5)$$

For $\tau > 1/2$, this gives an in-support optimistic bootstrap target without explicit action maximization. LWD keeps this asymmetric-bootstrap principle, but replaces scalar *expectile* regression with a distributional value model and *quantile*-based extraction.

### C. Flow Matching and Q-learning with Adjoint Matching

Flow Matching (FM) [33] represents a generative policy as a time-dependent vector field. Given a data action chunk $\mathbf{a}^1 = \mathbf{a}$ and Gaussian noise $\mathbf{a}^0 \sim \mathcal{N}(0, I)$, FM defines the interpolation

$$\mathbf{a}^w = (1 - w)\mathbf{a}^0 + w\mathbf{a}^1, \qquad w \in [0, 1], \quad (6)$$

and trains a conditional vector field $f_\theta(s, \mathbf{a}^w, w)$ to match the velocity $\mathbf{a}^1 - \mathbf{a}^0$. Flow-based VLA policies use this construction as the action-generation head [5, 6]. Q-learning with Adjoint Matching (QAM) [30] uses critic gradients for policy extraction without backpropagating through the full denoising trajectory. Given a pretrained reference flow $f_\beta$ and a critic $Q_\phi$, QAM defines the KL-regularized improvement target $\pi^*(\mathbf{a} \mid s) \propto \pi_\beta(\mathbf{a} \mid s) \exp\left(Q_\phi(s, \mathbf{a})/\lambda\right)$, where $\lambda$ is the temperature. The resulting update can be written as a local regression objective along trajectories of the reference flow:

$$\Delta f_{\theta,\beta} = f_\theta(s, \mathbf{a}^w, w) - f_\beta(s, \mathbf{a}^w, w),$$

$$\mathcal{L}_{\text{QAM}}(\theta) = \mathbb{E}\left[\int_0^1 \left\|\frac{2\Delta f_{\theta,\beta}}{\sigma_w} + \sigma_w \tilde{g}_w\right\|_2^2 \mathrm{d}w\right]. \quad (7)$$

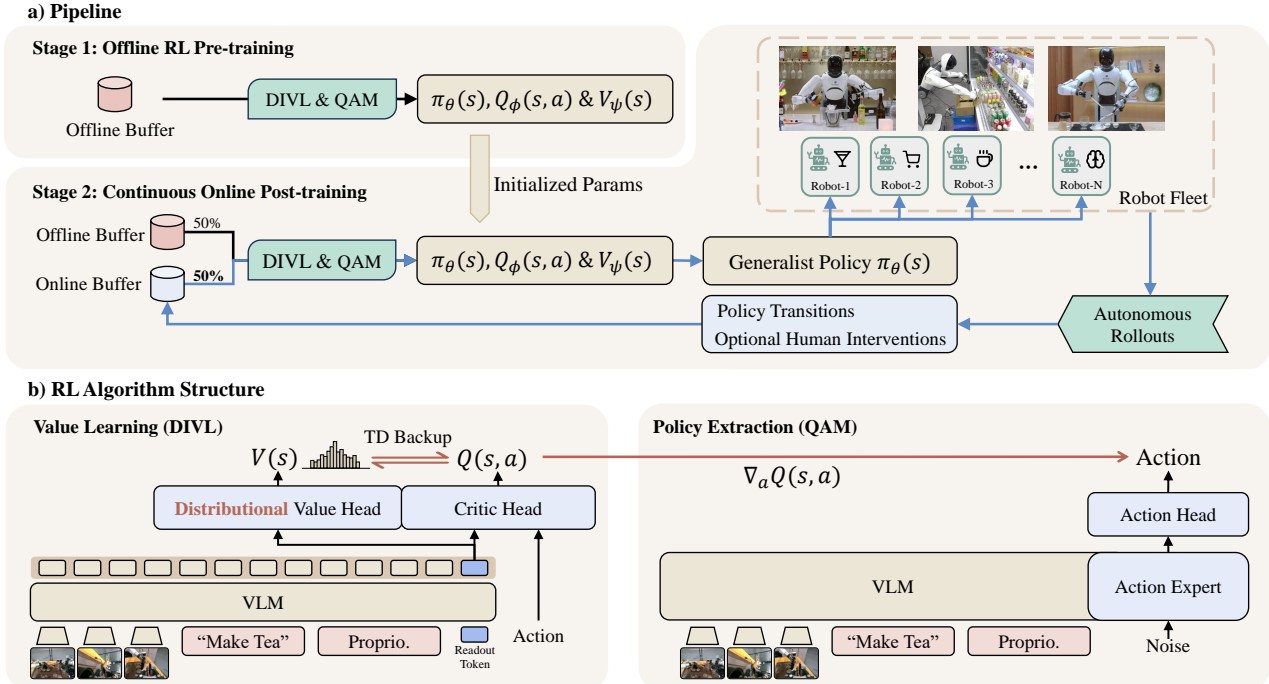

Fig. 2. **LWD overview.** (a) Offline pretraining initializes the learner on $\mathcal{B}_{\text{off}}$, then fleet deployment continually adds online replay for mixed offline-online updates. (b) The policy is optimized with QAM, while the critic and distributional value model are trained with TD learning under DIVL.

where $\sigma_w = \sqrt{2(1-w)w}$ and $\tilde{g}_w$ is the adjoint state with terminal condition

$$\tilde{g}_1 = -\nabla_{\mathbf{a}}\left[Q_\phi(s, \mathbf{a}^1)/\lambda\right]. \tag{8}$$

LWD uses QAM [30] as its policy-extraction mechanism, using the DIVL critic to form local regression targets for the flow policy.

## IV. LEARNING WHILE DEPLOYING

LWD follows the two-stage offline-to-online procedure in Fig. 2(a). The offline stage trains the policy, critic, and distributional value model on a static buffer $\mathcal{B}_{\text{off}}$. The online stage deploys the current policy across the fleet, appends autonomous rollouts and optional intervention segments to $\mathcal{B}_{\text{on}}$, updates $V_\psi$, $Q_\phi$, and $\pi_\theta$ on mixed replay from $\mathcal{B}_{\text{off}} \cup \mathcal{B}_{\text{on}}$, and periodically republishes updated checkpoints. This closes the deployment-improvement loop.

The learner-side optimization has two key components. *Distributional Implicit Value Learning (DIVL)* trains the critic $Q_\phi$ and distributional value model $V_\psi$. *QAM*-based policy extraction updates the flow policy $\pi_\theta$ using the action gradient of $Q_\phi$ learned from DIVL.

### A. Distributional Implicit Value Learning

Distributional Implicit Value Learning (DIVL) is the value-learning component of LWD. It models the state-conditioned distribution of replay action-values and uses a $\tau$-quantile of that distribution as the bootstrap statistic for the chunk-level critic $Q_\phi(s_t, \mathbf{a}_t)$. This keeps the asymmetric value-learning

principle of IQL [21] while avoiding a single scalar expectile target.

Concretely, the distributional value model $V_\psi(s_t)$ represents the state-conditioned distribution of dataset action-values [4]:

$$p_\psi(v \mid s_t) = P(v = Q_\phi(s_t, \mathbf{a}_t) \mid \mathbf{a}_t \sim \mathcal{D}(\cdot \mid s_t)), \tag{9}$$

where $\mathcal{D}(\cdot \mid s_t)$ denotes the empirical replay action distribution conditioned on $s_t$. Thus, $V_\psi(s_t)$ represents a distribution of critic values over replay actions rather than a single scalar estimate. We fit this distribution by minimizing the negative log-likelihood of scalar EMA-critic targets:

$$\mathcal{L}_V(\psi) = \mathbb{E}_{(s_t, \mathbf{a}_t) \sim \mathcal{D}}\left[-\log p_\psi\big(Q_{\bar{\phi}}(s_t, \mathbf{a}_t) \mid s_t\big)\right]. \tag{10}$$

We implement $p_\psi$ categorically; Appendix A1 gives the discretization and projection details. Compared with scalar regression, the distributional parameterization better preserves heterogeneous replay outcomes [22] and supports both quantile extraction and uncertainty-aware adaptation of $\tau$.

We use the $\tau$-quantile of $V_\psi(s_t)$ as the bootstrap statistic:

$$\text{Quant}_\tau\big(V_\psi(s_t)\big) \triangleq \inf\left\{v : F_\psi(v \mid s_t) \geq \tau\right\}, \tag{11}$$

where $F_\psi(v \mid s_t)$ is the cumulative distribution function induced by $p_\psi$. This yields the TD target

$$y_Q = \mathbf{r}_t + \gamma^H \text{Quant}_\tau\big(V_\psi(s_{t+H})\big), \tag{12}$$

and the critic loss

$$\mathcal{L}_Q(\phi) = \mathbb{E}_{(s_t, \mathbf{a}_t, \mathbf{r}_t, s_{t+H}) \sim \mathcal{D}}\left[\big(Q_\phi(s_t, \mathbf{a}_t) - y_Q\big)^2\right]. \tag{13}$$

The $\tau$-quantile is an in-distribution optimistic bootstrap statistic over replay actions rather than an explicit max backup over the full action space. This favors high-value replay actions without aggressive extrapolation beyond the data. IQL addresses the same issue with scalar expectile value regression; DIVL keeps the same asymmetric value-learning principle through a distributional model and quantile statistic.

To make this connection explicit, we write the value target under a generalized asymmetric loss family $\rho_{\tau,p}(u) = |\tau - \mathbb{I}(u < 0)| \cdot |u|^p$, where $p = 2$ gives the expectile form used by IQL and $p = 1$ gives the quantile form used by DIVL. Indeed, DIVL's two-step procedure of distributional value learning and statistic extraction has the same optimum as IQL's direct asymmetric value regression objective under the same asymmetric loss. The theoretical analysis is provided in Appendix A2.

The value of $\tau$ controls bootstrap optimism. In mixed-task replay, we adapt it using uncertainty in the learned value distribution. Specifically, we use the normalized entropy of the categorical distribution $p_\psi(\cdot \mid s)$ as the uncertainty signal:

$$\mathcal{H}(s) = -\tfrac{1}{\log C} \sum_{c=1}^{C} p_{\psi,c}(s) \log p_{\psi,c}(s), \qquad (14)$$

where $C$ is the number of categories and $p_{\psi,c}(s)$ is the probability assigned to category $c$. The adaptive schedule is

$$\tau(s) = \mathrm{clip}\big(\tau_{\mathrm{base}} - \alpha\, \mathcal{H}(s),\ \tau_{\min},\ \tau_{\max}\big), \qquad (15)$$

where $\tau_{\mathrm{base}}$ is the target for confident states, $\alpha \geq 0$ controls uncertainty sensitivity, and hyperparameter values are reported in Appendix B2. We treat $\tau(s)$ as stop-gradient in the TD target.

### B. Policy Extraction via QAM

Policy extraction in LWD starts from a pretrained flow-matching VLA. Likelihood-based actor updates [47, 43, 21, 59] are poorly matched to flow-based action generation because they require evaluating the log likelihood of action chunks under the multi-step denoising process. Direct critic back-propagation through the full multi-step generation process is computationally expensive and numerically unstable for flow policies (Appendix A3).

We therefore use QAM for policy extraction [30] as shown in Fig. 2(b). QAM reformulates trajectory-level policy optimization into a local regression objective along the reference flow. The DIVL critic $Q_\phi$ supplies the terminal action gradient to initialize $\tilde{g}_1$ (Eq. (8)), while $\pi_\beta$ remains fixed as the behavior-cloned reference flow and $\pi_\theta$ is optimized throughout offline and online training. For each state $s_t$, we sample Gaussian noise $\mathbf{a}_t^0$, generate flow trajectories via $\pi_\theta$, evaluate $\nabla_\mathbf{a} Q_\phi(s_t, \mathbf{a}_t^1)$ at the endpoint, and regress $\pi_\theta$ by minimizing Eq. (7).

### C. Offline to Online RL Training Pipeline

Following the LWD loop in Fig. 2(a), post-training proceeds in two stages that share the same value-learning and policy-extraction objectives but differ in data source. Algorithms are summarized in Appendix A4.

The offline stage trains on an offline buffer $\mathcal{B}_{\mathrm{off}}$ containing demonstrations, historical rollouts, and play data. Terminal success or failure labels assign sparse binary rewards, and the resulting replay is used to pre-train $\pi_\theta$, $Q_\phi$, and $V_\psi$ before deployment.

Moreover, since long-horizon tasks last thousands of steps and have extremely sparse rewards, the one-step target in Eq. (12) can propagate success signals slowly. We therefore use an $n$-step chunk-level TD target in the offline stage to cold-start the critic and distributional value model:

$$y_Q = \sum_{i=0}^{n-1} \gamma^{iH} \mathbf{r}_{t+iH} + \gamma^{nH} \mathrm{Quant}_{\tau(s_{t+nH})}\big(V_\psi(s_{t+nH})\big), \qquad (16)$$

where $n = 1$ for short-horizon tasks and $n = 10$ for long-horizon tasks. This target accelerates sparse reward propagation through the fixed offline replay buffer. During online training, we instead use 1-step chunk-level TD targets, since longer backups are more likely to cross mixed segments from policy and intervention.

The online stage deploys the offline-initialized policy to the fleet and appends autonomous policy transitions and human intervention segments to $\mathcal{B}_{\mathrm{on}}$. Training then continues with the same value-learning and policy-extraction objectives on mixed replay from $\mathcal{B}_{\mathrm{off}} \cup \mathcal{B}_{\mathrm{on}}$, while updated policy checkpoints are periodically published back to the robots.

## V. EXPERIMENTAL EVALUATIONS

We evaluate LWD on eight real-world manipulation tasks: four grocery restocking tasks and four long-horizon tasks. The experiments ask whether deployment-time online updates from a shared robot fleet improve over static or offline policies, how LWD compares with representative post-training baselines, and whether the DIVL value-learning design contributes to the gains.

### A. Experimental Setup

*a) Tasks and metrics.:* The grocery tasks include flat-shelf restocking, misplaced-item correction, freezer restocking with door operation, and open-cooler restocking with carton handling, as shown in Fig. 3. These tasks evaluate language-conditioned object selection, clutter handling, and placement under varied store layouts. The long-horizon tasks include brewing Gongfu Tea, making Fruit Juice, making Cocktail, and packing shoes into a shoebox. Each long-horizon episode lasts 3–5 minutes and contains 5–8 annotated sub-steps involving contact-rich skills such as grasp adjustment, pouring, tool use, container handling, and recovery.

We report binary task success for grocery restocking tasks, following SOP [45]. For long-horizon tasks, we report a step-wise success score: each annotated sub-step is scored as 1 for autonomous success, 0.5 for minor imperfection or a single retry, and 0 for failure after multiple attempts; the task score averages over sub-steps. Detailed scoring rubrics are provided in Appendix C1.

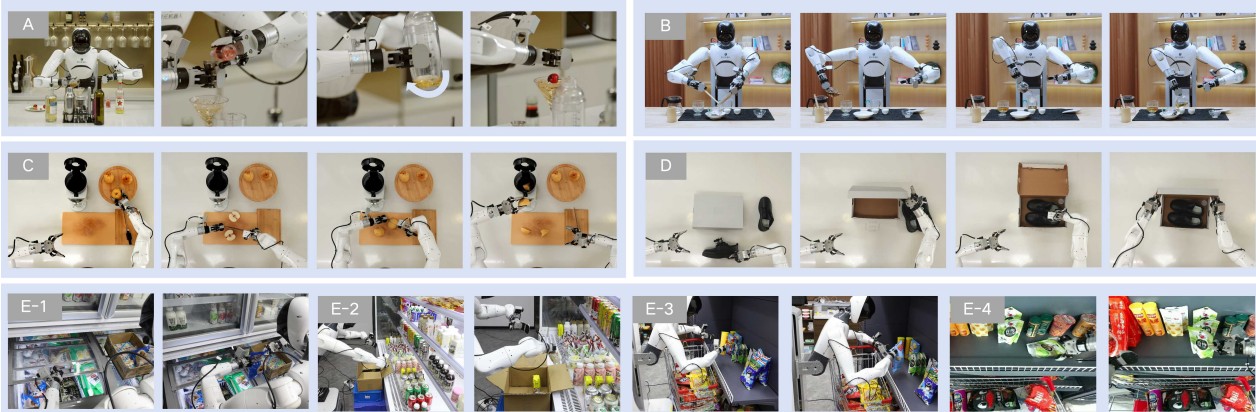

Fig. 3. **Evaluation tasks.** Panels A–D show the long-horizon tasks: Make Cocktail, Brew Gongfu Tea, Make Fruit Juice, and Pack Shoes. Panel E summarizes the four grocery restocking tasks.

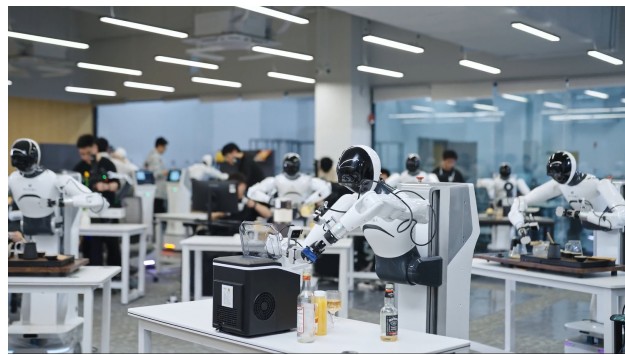

Fig. 4. **Robot fleet training.** Sixteen Agibot G1 robots collect online rollouts for LWD.

*b) Robot fleet and online protocol.:* All experiments use the Agibot G1 dual-arm manipulation platform with three RGB cameras and 30 Hz joint-position control. We deploy 16 robots for asynchronous online rollout collection: 4 robots for grocery restocking and 3 robots for each long-horizon task shown in Fig. 4. Each online method receives a 4-hour wall-clock budget, corresponding to approximately 60 hours of robot data. Episodes from all tasks are pooled into one online replay buffer, mixed with offline replay during training, and the centralized learner broadcasts the updated shared policy to the fleet every 50 training steps. Hyperparameter details and system infrastructure are provided in Appendix B2 and Appendix D.

*c) Baselines.:* All post-training methods start from the same SFT reference policy, which fine-tunes the pretrained VLA policy on human demonstrations using the standard flow-matching imitation loss [33]. RECAP [2] uses autonomous rollout data and advantage-conditioned policy improvement. HG-DAgger [19] uses online rollout data with human intervention segments. LWD (Offline) trains on the static offline replay buffer containing demonstrations, failed historical rollouts, and play data; LWD (Online) initializes from LWD (Offline) and continues on mixed offline-online replay with sparse

rewards. Implementation details and data budgets are given in Appendix C2.

### B. Main Results

Fig. 5 reports the quantitative comparison across all eight tasks, and Table I provides the complete per-task results. LWD (Online) achieves the best score, outperforming SFT, RECAP, HG-DAgger, and LWD (Offline) on all tasks. LWD also reduces mean long-horizon cycle time by 23.75 seconds relative to SFT. Value-curve visualizations are shown in Appendix Figs. 9 and 8.

The advantage of LWD is most pronounced on long-horizon tasks, where LWD (Online) reaches an average step-wise score of 0.91, compared with 0.68 for SFT, 0.77 for RECAP, 0.73 for HG-DAgger, and 0.79 for LWD (Offline). These tasks contain sparse terminal rewards, long chains of dependent sub-skills, and many recoverable failure modes. LWD uses both successful and failed trajectories for reward-based policy improvement, allowing terminal outcomes to propagate to earlier decisions through TD backups and improving recovery behavior during deployment.

On grocery restocking tasks, most post-training methods already achieve high scores, leaving less room for improvement. Even in this saturated regime, LWD (Online) remains at or near the best result on every grocery task, suggesting that online fleet learning improves long-horizon performance without degrading the shorter semantic manipulation tasks.

### C. Ablation Study

We compare DIVL with scalar expectile value regression while keeping the other training setup fixed. DIVL improves over the scalar baseline on both task groups, with larger gains on long-horizon tasks: 9.7% in the offline stage and 16.7% in the online stage. This is consistent with the intuition that distributional value learning preserves heterogeneous outcomes in replay, including rare high-return continuations, whereas a scalar value compresses them into a single expectation.

We further ablate the adaptive $\tau$ strategy used in DIVL during offline LWD training. We compare the adaptive schedule against a constant-$\tau$ baseline, where the constant value ($\tau = 0.52$)

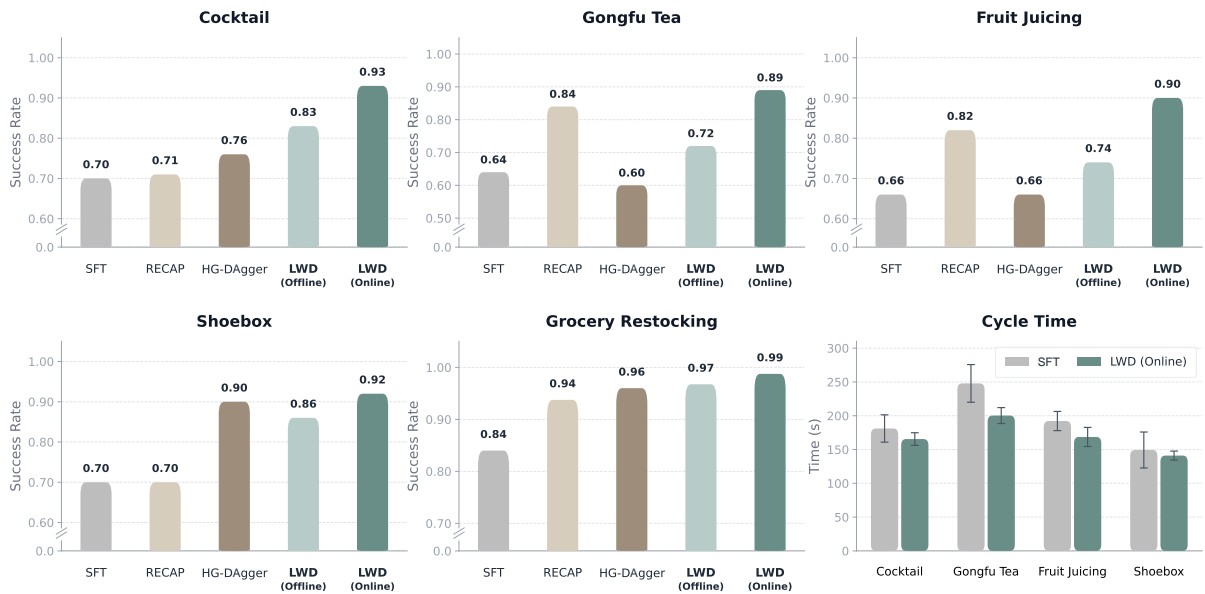

Fig. 5. **Quantitative results.** LWD improves long-horizon scores and reduces mean cycle time relative to the static SFT reference policy. Complete per-task results are shown in Table I.

TABLE I
**RESULTS ON EIGHT REAL-WORLD MANIPULATION TASKS.** GROCERY TASKS USE BINARY SUCCESS; LONG-HORIZON TASKS USE AVERAGE STEP-WISE
SCORE. THE BEST RESULT PER COLUMN IS BOLD.

| Method | Grocery Restocking Tasks | | | | Long-Horizon Tasks | | | | Average |
|---|---|---|---|---|---|---|---|---|---|
| | Restocking | Correction | Freezer | Open-Cooler | Gongfu Tea | Fruit Juice | Cocktail | Shoebox | |
| SFT [33] | 0.70 | 0.88 | 0.83 | 0.95 | 0.64 | 0.66 | 0.70 | 0.70 | 0.76 |
| RECAP [2] | 0.95 | 0.96 | 0.94 | 0.95 | 0.84 | 0.82 | 0.71 | 0.70 | 0.85 |
| HG-DAgger [19] | **1.00** | 0.92 | 0.92 | **1.00** | 0.60 | 0.66 | 0.76 | 0.90 | 0.85 |
| LWD (Offline, Ours) | **1.00** | **1.00** | 0.92 | 0.95 | 0.72 | 0.74 | 0.83 | 0.86 | 0.88 |
| **LWD (Online, Ours)** | **1.00** | **1.00** | **0.97** | 0.98 | **0.89** | **0.90** | **0.93** | **0.92** | **0.95** |

TABLE II
**DIVL ABLATION.** AVERAGE SCORES BY TASK GROUP. COMPLETE
PER-TASK RESULTS ARE REPORTED IN APPENDIX TABLE IV.

| Value | Short | | Long | |
|---|---|---|---|---|
| | Off. | On. | Off. | On. |
| IQL | 0.96 | 0.97 | 0.72 | 0.78 |
| DIVL | 0.97 +1.0% | 0.99 +2.1% | 0.79 +9.7% | 0.91 +16.7% |

is set to the average $\tau$ observed from training statistics in the adaptive-$\tau$ run. Adaptive $\tau$ improves the offline average score from 0.84 to 0.88 (Appendix Table V). This indicates that conditioning $\tau$ on distributional entropy helps calibrate bootstrap optimism, making targets more conservative under high uncertainty and more optimistic when the value estimate is confident.

## VI. CONCLUSION

We present Learning While Deploying (LWD), a large-scale real-world reinforcement learning framework for post-training generalist robot policies. LWD first initializes the policy from previously collected robot data, then continues improving it

through online RL during deployment. The framework uses DIVL for value learning and QAM for policy extraction. Across eight real-world manipulation tasks spanning grocery restocking and long-horizon manipulation, LWD delivers the best overall performance, with the most pronounced improvements on long-horizon tasks.

These results suggest a practical path toward large-scale real-world deployment of continuously improving robot systems. With LWD, deployment is not only the setting in which the policy is evaluated, but also the mechanism through which the policy improves. Interaction data collected from the robot fleet is aggregated into a shared learning process, enabling a generalist policy to continue improving across tasks. This is critical for real-world robotic systems that must operate in heterogeneous tasks and environments.

Our method has several limitations. First, the current online learning pipeline updates with a straightforward real-time schedule. This design may not be optimal for larger-scale deployment or long-term continual improvement. More efficient and stable update strategies remain an important direction for future work. Second, our long-horizon experiments rely on a single short language instruction for each task. However,

complex tasks require stronger vision-language reasoning for task decomposition, as well as finer-grained prompts for closed-loop execution and error recovery. Third, our current policy learning framework does not explicitly model execution safety. Incorporating safety-aware learning and control mechanisms will be important for reliable real-world deployment. Despite these limitations, this work represents a step toward large-scale real-world deployment, with the long-term goal of continuously scaling robot learning systems for robust execution in unstructured environments.

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

## APPENDIX

### A. Additional Method Details

*1) Discretization of Distributional Value Model:* We instantiate the distributional value model $V_\psi(s)$ with a fixed categorical support $\{V_c\}_{c=1}^C$ spanning $[v_{\min}, v_{\max}]$. In our real-robot experiments, we use $C = 201$ atoms over $[-0.1, 1.1]$. The value head predicts logits over this support,

$$p_\psi(c \mid s) = \text{softmax}(V_\psi(s))_c, \qquad c \in \{1, \dots, C\}. \quad (17)$$

For each replay sample $(s, \mathbf{a})$, the scalar target $Q_{\bar\phi}(s, \mathbf{a})$ is clipped to $[v_{\min}, v_{\max}]$ and linearly projected onto the two neighboring atoms following the C51 projection [4]. This yields a target distribution $m(s, \mathbf{a})$ over atoms, and the distributional value model is trained by cross entropy:

$$\mathcal{L}_V(\psi) = -\mathbb{E}_{(s,\mathbf{a})\sim\mathcal{D}} \left[ \sum_{c=1}^C m_c(s, \mathbf{a}) \log p_\psi(c \mid s) \right]. \quad (18)$$

The discrete CDF is

$$F_\psi(V_c \mid s) = \sum_{i \le c} p_\psi(i \mid s), \quad (19)$$

and the quantile used in the DIVL TD target is obtained by selecting the first atom whose cumulative probability exceeds the desired level:

$$\text{Quant}_\tau(V_\psi(s)) = V_{\min\{c : F_\psi(V_c \mid s) \ge \tau\}}. \quad (20)$$

The normalized entropy used in the adaptive $\tau$ strategy is

$$\mathcal{H}(s) = -\frac{1}{\log C} \sum_{c=1}^C p_\psi(c \mid s) \log p_\psi(c \mid s) \in [0, 1]. \quad (21)$$

*2) Theoretical Analysis of Distributional View of Asymmetric Value Estimation:*

**Proposition 1** (Distributional view of asymmetric value learning)**.** *For any fixed asymmetric loss, direct scalar regression and our two-step procedure of fitting the value distribution and extracting the corresponding asymmetric statistic yield the same optimal scalar value.*

*Proof:* The goal is to show that, under idealized conditions, direct asymmetric optimization over dataset action-values and the two-step procedure of first fitting the state-conditioned distribution of dataset $Q$-values and then extracting the corresponding asymmetric statistic yield the same optimal scalar value.

Define the generalized asymmetric $L_p$ loss

$$\rho_{\tau,p}(u) = |\tau - \mathbb{I}(u < 0)| \cdot |u|^p, \quad (22)$$

where $\tau \in (0, 1)$ is the asymmetry parameter. In standard IQL, the scalar value is obtained by directly minimizing

$$J_{\text{direct}}(v) = \mathbb{E}_{\mathbf{a}\sim\mathcal{D}(\cdot\mid s)} \left[ \rho_{\tau,p}(Q(s, \mathbf{a}) - v) \right]. \quad (23)$$

The first-order optimality condition is

$$\frac{\mathrm{d}}{\mathrm{d}v} J_{\text{direct}}(v) = \int \mathcal{D}(\mathbf{a} \mid s) \cdot \frac{\mathrm{d}}{\mathrm{d}v} \rho_{\tau,p}(Q(s, \mathbf{a}) - v) \, \mathrm{d}\mathbf{a} = 0. \quad (24)$$

Now consider DIVL in the idealized limit of infinitely fine discretization and sufficient model capacity. Let $p_\psi(z \mid s)$ denote the learned state-conditioned density over dataset $Q$-values. At optimum, the cross-entropy objective recovers the pushforward distribution induced by $\mathbf{a} \sim \mathcal{D}(\cdot \mid s)$ through the mapping $v = Q(s, \mathbf{a})$:

$$p_\psi(v \mid s) = P(v = Q(s, \mathbf{a}) \mid \mathbf{a} \sim \mathcal{D}(\cdot \mid s)). \quad (25)$$

Thus, for any integrable test function $f(z)$,

$$\mathbb{E}_{v\sim p_\psi(\cdot\mid s)}[f(z)] = \mathbb{E}_{\mathbf{a}\sim\mathcal{D}(\cdot\mid s)}[f(Q(s, \mathbf{a}))]. \quad (26)$$

The second step of DIVL extracts a scalar statistic by minimizing

$$J_{\text{dist}}(v) = \mathbb{E}_{u\sim p_\psi(\cdot\mid s)} \left[ \rho_{\tau,p}(u - v) \right]. \quad (27)$$

Its first-order optimality condition is

$$\frac{\mathrm{d}}{\mathrm{d}v} J_{\text{dist}}(v) = \int p_\psi(u \mid s) \cdot \frac{\mathrm{d}}{\mathrm{d}v} \rho_{\tau,p}(u - v) \, \mathrm{d}u = 0. \quad (28)$$

Because $p_\psi(\cdot \mid s)$ is exactly the pushforward of $\mathbf{a} \sim \mathcal{D}(\cdot \mid s)$ under the random variable $u = Q(s, \mathbf{a})$, the above integral is identical to the direct objective's optimality condition after

change of variables. Therefore, $J_{\mathrm{direct}}$ and $J_{\mathrm{dist}}$ admit the same minimizer $v^*$ under the stated idealized assumptions. ∎

This establishes that direct asymmetric optimization and the distribution-fit-then-extract procedure are equivalent in the limit. In particular, $p = 2$ recovers the expectile statistic used in standard IQL, while $p = 1$ recovers the quantile statistic used by DIVL.

*3) Analysis of Direct Backpropagation for Flow-Based Policy:* Consider a flow-based policy that generates an action $x = x_1$ by integrating the vector field $\mathrm{d}x_t = f_\theta(x_t, t)$ from $t = 0$ to $1$ starting from $x_0 \sim \mathcal{N}$. Writing $x_1 = x_1(x_0; \theta)$ for the terminal sample induced by the flow, the standard RL objective for reward fine-tuning is

$$J(\theta) = \mathbb{E}_{x_0 \sim \mathcal{N}}\left[R\big(x_1(x_0; \theta)\big)\right], \qquad (29)$$

and a vanilla policy gradient requires differentiating through the entire ODE trajectory:

$$\nabla_\theta J(\theta) = \mathbb{E}_{x_0 \sim \mathcal{N}}\left[\nabla_x R(x_1) \cdot \int_0^1 \Phi(1, t)\frac{\partial f_\theta(x_t, t)}{\partial \theta} dt\right], \qquad (30)$$

where $\Phi(1, t) = \frac{\partial x_1}{\partial x_t}$ is the sensitivity matrix along the flow. In practice, this formulation is computationally expensive and numerically fragile because it requires backpropagation through the full ODE solver [11]. Adjoint Matching (Section IV-B) avoids this issue by reformulating trajectory-level optimization as local regression targets along the flow path.

*4) Algorithms:* Algorithm 1 outlines our two-stage offline-to-online training pipeline.

Algorithm 2 details the corresponding learner update step, consisting of DIVL and QAM.

*5) Architectures:* Fig. 2(b) shows the concrete neural network architecture used by LWD. The policy and value/critic networks are separate modules, isolating action generation from value and critic optimization. Only the policy checkpoint is asynchronously distributed to the robot fleet for inference, while the value and critic networks remain on the centralized learner.

We implement $V_\psi$ and $Q_\phi$ with a shared Gemma 3–SigLIP VLM backbone and separate prediction heads. The Gemma 3 language module and SigLIP vision encoder are initialized from publicly released Gemma 3-270M-IT [51] and SigLIP-So400M checkpoints [58], while the visual projection layer and value/critic heads are initialized from scratch.

Following the use of readout tokens as compact transformer representations [12, 48, 29], we apply the shared backbone to the multimodal sequence for state $s_t$ and denote the final hidden state of the readout token by $z_t$, which serves as the state representation for both value and critic prediction. The value head predicts logits over a fixed categorical support. Following the C51 projection [4], the scalar supervision target $Q_{\bar{\phi}}(s_t, \mathbf{a}_t)$ is clipped to the value support and linearly projected onto its two neighboring atoms, yielding a target distribution $m_t$.

The critic conditions on both the state representation $z_t$ and the action chunk $\mathbf{a}_t$. The action chunk is encoded with a learned temporal attention pooling layer and concatenated with

---

**Algorithm 1** LWD: Offline-to-Online Training Pipeline
___
**Require:** offline buffer $\mathcal{B}_{\mathrm{off}}$; demonstration dataset $\mathcal{D}_{\mathrm{demo}} \subset \mathcal{B}_{\mathrm{off}}$; online buffer $\mathcal{B}_{\mathrm{on}}$; robot actor fleet $\mathcal{F}$; offline budget $N_{\mathrm{off}}$; online budget $N_{\mathrm{on}}$; actor-sync period $N_{\mathrm{sync}}$.
1: Pretrain policy $\pi_\theta \leftarrow \mathcal{D}_{\mathrm{demo}}$
2: Set fixed reference policy $\pi_\beta \leftarrow \pi_\theta$
3: Initialize $Q_\phi$, $V_\psi$; set target $Q_{\bar{\phi}} \leftarrow Q_\phi$
    `// Stage 1: Offline Pretraining`
4: **for** $i \leftarrow 1 : N_{\mathrm{off}}$ **do**
5:     Sample mini-batch $\mathcal{B}^{\mathrm{mini}} \sim \mathcal{B}_{\mathrm{off}}$
6:     $(Q_\phi, V_\psi, \pi_\theta, Q_{\bar{\phi}}) \leftarrow \text{LEARNER}(\mathcal{B}^{\mathrm{mini}}; Q_\phi, V_\psi, \pi_\theta, \pi_\beta, Q_{\bar{\phi}})$
7: **end for**
    `// Stage 2: Continuous Online Training`
    **Robot actor process (Asynchronously):**
8: Deploy $\pi_\theta$ to each robot from $\mathcal{F}$
9: **while** online training is active **do**
10:     done $\leftarrow False$; $T \leftarrow 0$
11:     **while** not done **do**
12:         Execute $\mathbf{a} \leftarrow \pi_\theta(s)$ until done
13:         **if** intervention is required **then**
14:             Human intervenes: $\mathbf{a} \leftarrow \mathbf{a}_H$
15:         **end if**
16:         $s' \leftarrow UpdateObs(s, \mathbf{a})$
17:         done $\leftarrow \mathbb{I}[TimeLimit \vee Failure \vee Success]$
18:         $r \leftarrow \mathbb{I}[\text{done} \wedge Success]$
19:         $T \leftarrow T + 1$
20:     **end while**
21:     $\mathbf{r} \leftarrow UpdateChunkedReward(r)$
22:     $\mathcal{B}_{\mathrm{on}} \leftarrow \mathcal{B}_{\mathrm{on}} \cup \{(s_t, \mathbf{a}_t, \mathbf{r}_t, s'_{t+H})\}$
23:     $\pi_\theta \leftarrow FetchNewPolicy(\pi_\theta^{new})$
24: **end while**
    **Central learner process (Asynchronously):**
25: **for** $j \leftarrow 1 : N_{\mathrm{on}}$ **do**
26:     Sample mini-batch $\mathcal{B}^{\mathrm{mini}} \sim \{\mathcal{B}_{\mathrm{off}} \cup \mathcal{B}_{\mathrm{on}}\}$
27:     $(Q_\phi, V_\psi, \pi_\theta, Q_{\bar{\phi}}) \leftarrow \text{LEARNER}(\mathcal{B}^{\mathrm{mini}}; Q_\phi, V_\psi, \pi_\theta, \pi_\beta, Q_{\bar{\phi}})$
28:     **if** $j \bmod N_{\mathrm{sync}} = 0$ **then**
29:         Deploy latest policy $\pi_\theta$ to each robot from $\mathcal{F}$
30:     **end if**
31: **end for**
32: **return** $Q_\phi$, $V_\psi$, $\pi_\theta$

---

$z_t$. The resulting representation is fed into two scalar critic heads in a clipped double-Q design, where the minimum critic estimate is used for DIVL target construction and TD backups to mitigate overestimation.

The actor follows the $\pi_{0.5}$ flow-based VLA architecture [6]. It consists of a PaliGemma vision-language backbone, instantiated with a Gemma-2B language model and a SigLIP vision encoder, together with a Gemma-300M action expert for flow-based action generation.

In the offline RL stage, both the actor and the value/critic networks are fully fine-tuned; the resulting weights initialize online training. During online QAM updates, the policy VLM

**Algorithm 2** LEARNER: Single Update of DIVL and QAM

---

**Require:** mini-batch $\mathcal{B}^{\mathrm{mini}} = \{(s_t, \mathbf{a}_t, \mathbf{r}_t, s_{t+H})\}$; critic $Q_\phi$ with target $Q_{\bar{\phi}}$; distributional value $V_\psi$; policy $\pi_\theta$ with reference policy $\pi_\beta$; EMA rate $\rho$.
   // Distributional Implicit Value Learning
1: Update $\psi$ by minimizing Eq. (10)
2: Compute TD target $y_Q$ via Eq. (16)
3: Update $\phi$ by minimizing Eq. (13)
4: $\bar{\phi} \leftarrow (1 - \rho)\,\bar{\phi} + \rho\,\phi$
   // Policy Extraction via QAM
5: Sample Gaussian noise $\mathbf{a}_t^0 \sim \mathcal{N}(0, I)$
6: Roll out the reference trajectory $\{\mathbf{a}_t^w\}_{w \in [0,1]}$ via $\pi_\theta$
7: Update $\theta$ by minimizing Eq. (7) with $\tilde{g}_1$ set from action gradient $\nabla_\mathbf{a} Q_\phi(s, \mathbf{a}_t^1)$ via Eq. (8)
8: **return** $(Q_\phi, V_\psi, \pi_\theta, Q_{\bar{\phi}})$

---

backbone is frozen and only the action expert is updated, while the value and critic networks continue to be fully fine-tuned on mixed replay. This design keeps online policy updates efficient and preserves the pretrained vision-language representations, while allowing the value and critic networks to adapt to the evolving replay distribution and provide updated policy-improvement signals.

### B. Implementation and Training Details

*1) Offline Data:* The offline buffer $\mathcal{B}_{\mathrm{off}}$ consists of three types of data: *demonstration* data collected by human experts, *rollout* data produced by historical policies during prior evaluations, and *play* data in which a human operator explores failure modes and edge cases. Demonstrations are successful trajectories, rollouts contain both successes and failures, and play data is treated as unsuccessful exploratory data. Table III summarizes the data composition in hours by task. Fig. 6 shows the aggregate source distribution, illustrating the relative contribution of demonstrations, historical rollouts, and play data.

*2) Training Hyperparameters:* The policy emits action chunks with horizon $H = 30$. The policy is optimized with AdamW [37] using a base learning rate of $2 \times 10^{-5}$ and a cosine decay schedule. The value and critic networks are trained with Adam using a base learning rate of $5 \times 10^{-4}$, also with a cosine decay schedule.

For temporal-difference backups, we use $\gamma = 0.9999$. During offline training, we use $\tau_{\mathrm{base}} = 0.6$ and uncertainty-sensitivity coefficient $\alpha = 0.3$ for DIVL. During online training, we use $\tau_{\mathrm{base}} = 0.9$ and $\alpha = 0.3$. Adaptive $\tau$ is driven by the normalized entropy of the categorical value distribution: more diffuse states receive smaller $\tau$, while more confident states retain more optimistic targets. Target critic and value networks are updated with EMA rate 0.005, and the QAM policy-extraction temperature is $\lambda = 2$. The $\tau$ and entropy values during offline and online training are visualized in Fig. 7. Entropy decreases throughout offline-to-online training, indicating increasing

confidence in the value functions. Accordingly, the quantile parameter $\tau$ is increased, encouraging the policy to favor higher-value solutions.

For value learning, offline training uses 10-step chunk-level TD for long-horizon tasks and 1-step chunk-level TD for the grocery restocking tasks. Online training uses 1-step chunk-level TD for all tasks. During online training, each learner update samples mini-batches from $\mathcal{B}_{\mathrm{off}} \cup \mathcal{B}_{\mathrm{on}}$ with an approximately balanced ratio of 1:1.

*3) Checkpoint Initialization:* We first train an imitation-learning checkpoint by adapting the pretrained $\pi_{0.5}$ VLA policy on the demonstration data with behavior cloning. LWD (Offline) initializes its policy from this imitation-learning checkpoint, then trains the policy with the Adjoint Matching loss and trains the critic and distributional value model with DIVL. LWD (Online) initializes from the LWD (Offline) checkpoint, including both policy and value-learning modules, and continues training on mixed offline-online replay.

### C. Additional Experimental Details

*1) Evaluation Tasks, Metrics, and Cycle Time:* Fig. 3 illustrates the eight real-world manipulation tasks used in the main evaluation. The four grocery restocking tasks are flat-shelf restocking, misplaced-item correction, freezer restocking with door operation, and open-cooler restocking with carton handling. In each episode, the robot must identify the object specified by language instruction among cluttered candidates, handle shelf or container layout variation, and complete the required placement within the time limit.

The four long-horizon tasks are brewing Gongfu Tea, making Fruit Juice, making Cocktail, and packing shoes into a shoebox. Episodes typically last 3–5 minutes and include 5–8 annotated sub-steps. The annotated sub-steps correspond to task-level milestones such as adding, brewing, pouring, distributing, and serving tea; cutting fruit, transferring pieces, closing the juicer, and starting it; measuring, mixing, shaking, pouring, and garnishing a cocktail; and opening, packing, closing, and placing the shoebox. Evaluation episodes include natural reset variability in object poses, tool locations, ingredients, scene initialization, perturbations, and occasional retry or recovery situations.

For grocery restocking, an episode is successful if the robot follows the correct language instruction and completes the task within the time limit. For long-horizon tasks, trained human evaluators assign each sub-step a score of 1 for fully autonomous success, 0.5 for success with minor imperfection or a single retry, and 0 for failure after multiple attempts. Cycle time is computed over both successful and failed attempts, with failed trajectories clipped at predefined task-specific timeout thresholds. Fig. 5 reports the success-score and cycle-time comparison.

*2) Reference Policy and Baseline Implementations:* We obtain the reference policy by supervised fine-tuning [33] the pretrained $\pi_{0.5}$ VLA policy on 336.6 hours of demonstration data, as shown in Table III. The model is trained with a flow-matching loss, where the interpolated noisy action $\mathbf{a}^w$ is defined

TABLE III

OFFLINE DATA COMPOSITION (HOURS). DEMONSTRATIONS ARE EXPERT-COLLECTED SUCCESSFUL DATA; ROLLOUTS ARE GENERATED BY HISTORICAL POLICIES AND CONTAIN BOTH SUCCESSES AND FAILURES; PLAY DATA CONSISTS OF HUMAN-GUIDED EXPLORATIONS OF FAILURE MODES.

| Task | Demo | Rollout (Succ) | Rollout (Fail) | Play | Total |
|---|---|---|---|---|---|
| Restocking | 14.5 | 10.7 | 1.7 | 7.5 | 34.4 |
| Correction | 12.7 | 10.8 | 1.7 | 9.7 | 34.9 |
| Freezer | 10.8 | 7.7 | 4.6 | 3.3 | 26.4 |
| Open-Cooler | 11.1 | 13.7 | 1.3 | 0.8 | 26.9 |
| *Grocery Restocking Tasks (subtotal)* | *49.2* | *42.9* | *9.3* | *21.3* | *122.7* |
| Gongfu Tea | 102.3 | 12.4 | 4.4 | 43.6 | 162.7 |
| Fruit Juicing | 100.5 | 17.4 | 14.7 | 47.1 | 179.8 |
| Cocktail | 47.3 | 4.4 | 7.4 | 28.0 | 87.1 |
| Shoebox | 37.3 | 11.7 | 3.3 | 48.0 | 100.3 |
| *Long-Horizon Tasks (subtotal)* | *287.5* | *45.9* | *29.8* | *166.7* | *529.8* |
| **All Tasks** | **336.6** | **88.8** | **39.2** | **187.9** | **652.5** |

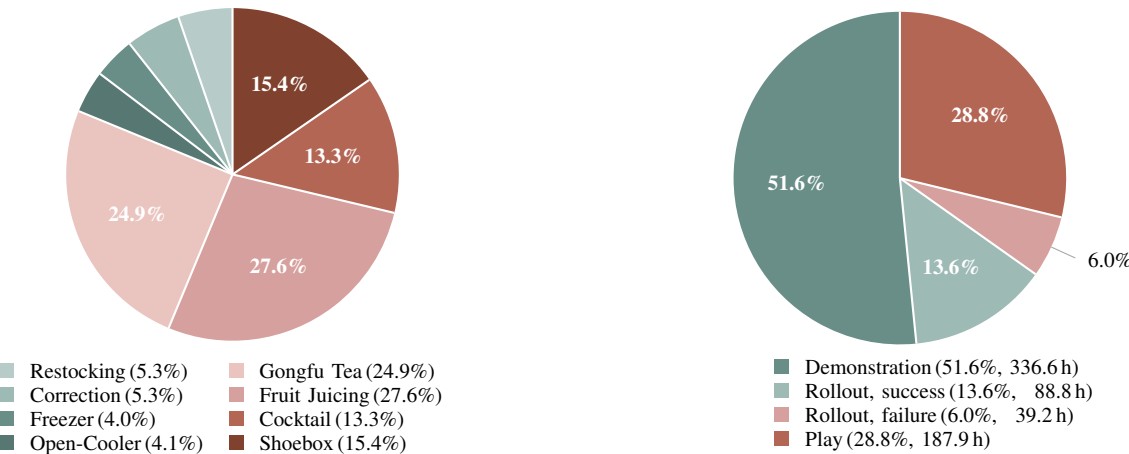

(a) By task: the Grocery Restocking tasks 18.8% | Long-Horizon 81.2%

Restocking (5.3%)   Gongfu Tea (24.9%)
Correction (5.3%)   Fruit Juicing (27.6%)
Freezer (4.0%)   Cocktail (13.3%)
Open-Cooler (4.1%)   Shoebox (15.4%)

(b) By source, colored by outcome: Successful 65.2% | Failure 34.8%

Demonstration (51.6%, 336.6 h)
Rollout, success (13.6%, 88.8 h)
Rollout, failure (6.0%, 39.2 h)
Play (28.8%, 187.9 h)

Fig. 6. **Offline data composition of the 652.5-hour buffer along two axes.** (a) Distribution across tasks: the grocery restocking tasks (green) and long-horizon tasks (red); long-horizon episodes dominate the buffer by volume due to their substantially longer duration. (b) Distribution across the three data sources—expert *demonstrations* (always successful), *rollouts* from historical policies (mixed success and failure outcomes), and human-guided failure-mode *play* (always unsuccessful). Wedges are colored by trajectory outcome so that the overall success/failure split across the buffer is directly legible: roughly one-third of the buffer is failure data, which the behavior-cloning baselines cannot use but which provides an informative learning signal for LWD. Per-task hours are reported in Table III.

in Eq. (6). The objective trains the conditional vector field $f_\theta(s, \mathbf{a}^w, w)$ to match the velocity $\mathbf{a}^1 - \mathbf{a}^0$:

$$\mathcal{L}_{\text{SFT}} = \mathbb{E}\left[\left\|f_\theta(s, \mathbf{a}^w, w) - (\mathbf{a}^1 - \mathbf{a}^0)\right\|_2^2\right]. \quad (31)$$

This reference policy is used for all the post-training methods.

For the RECAP [2] baseline, we initialize from the reference policy and adapt RECAP to the eight-task generalist setting. We collect two rounds of autonomous rollouts: Round 1 uses the SFT checkpoint, and Round 2 uses the RECAP checkpoint obtained after training on Round 1. Each round contains approximately 60 robot-hours pooled across all eight tasks. Following RECAP, we train a value model to compute advantage labels over the combined dataset of demonstrations and both autonomous rollout rounds; the value model uses the same value-network architecture as LWD, described in Section A5, but only the value head. We compute the lookahead

advantage and binary improvement label as

$$A(s_t, \mathbf{a}_t) = \sum_{t'=t}^{t+H-1} r_{t'} + V(s_{t+H}) - V(s_t), \quad (32)$$

$$I_t = \mathbb{1}[A^{\pi_{\text{ref}}}(s_t, \mathbf{a}_t) > \epsilon \ \vee \ c_t = 1], \quad (33)$$

where $c_t$ indicates that the transition is a human intervention or correction, which is treated as positive following RECAP when present. We use $H = 30$ as our action horizon length and select a single global advantage threshold $\epsilon$ so that 30% of transitions in the combined training set satisfy the positive-advantage condition. This threshold is selected from training data only and is shared across all tasks to avoid task-specific tuning. After the second rollout round, we train RECAP for one epoch over the combined dataset and evaluate the resulting checkpoint.

For the HG-DAgger [19] baseline, we initialize from the same reference policy checkpoint and run interactive imitation

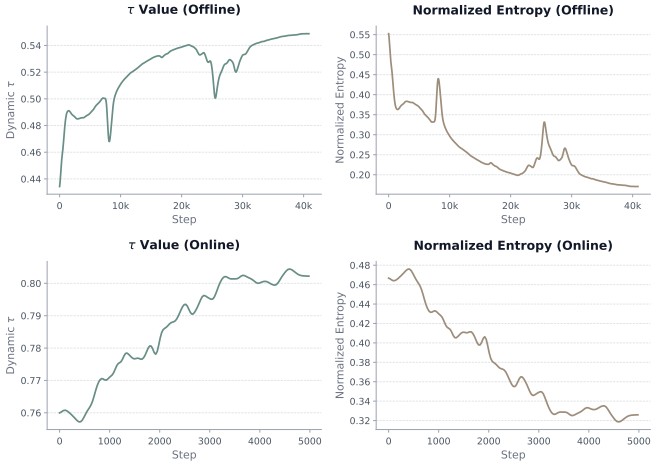

Fig. 7. **Dynamic $\tau$ and normalized entropy during offline-to-online training.** All curves are smoothed for readability. Entropy decreases throughout both stages, indicating increasing confidence in value estimation. Accordingly, $\tau$ is increased, leading to improved training performance.

learning on the eight-task suite. During online execution, human operators provide intervention segments when corrections are needed. These intervention segments are aggregated with autonomous rollouts to form an online training buffer of approximately 60 robot-hours pooled across all eight real-world tasks. The online buffer, together with the offline demonstration data buffer, is used for training HG-DAgger. We train HG-DAgger from the reference policy checkpoint using the same batch size and training-time budget as the corresponding online post-training runs, and evaluate the resulting checkpoint.

For a fair comparison, the post-training baselines use the same policy optimizer and learning-rate schedule as LWD.

*3) Complete Ablation Results:* Table IV reports the complete per-task results for the value-estimation ablation summarized in Section V. The comparison isolates the value-learning method by replacing DIVL with scalar expectile value regression while keeping the remaining training setup fixed.

Table V compares the adaptive $\tau$ schedule with a constant-$\tau$ baseline in offline LWD training. The constant value is set to the empirical average value from the adaptive-$\tau$ training statistics rather than tuned on evaluation performance. Adaptive $\tau$ improves the offline average score from $0.84$ to $0.88$, with especially clear gains on Restocking, Correction, and Cocktail. This is consistent with entropy-conditioned $\tau$ calibrating bootstrap optimism: uncertain value distributions use more conservative targets, while confident estimates can retain more optimistic targets.

*4) Complementary Qualitative Results of DIVL:* Fig. 8 visualizes the value estimate during a successful and a failed Gongfu Tea episode. In the successful episode, the value tends to increase as the robot completes key sub-steps and approaches task completion. In the failure episode, the value fluctuates locally but remains lower after execution stops making progress toward the annotated task milestones.

Fig. 9 visualizes the predicted value distributions for the same episodes. In the successful episode, the predicted distribution

remains unimodal, with its mode steadily increasing from approximately 0.4 to 1.0 as the task progresses. In contrast, the failure episode exhibits only marginal mode progression, increasing from approximately 0.5 to 0.6 before plateauing. These results indicate that the predicted value distribution provides a fine-grained signal to track policy progress and distinguish successful execution from failure cases.

### D. Distributed Data Infrastructure

Fig. 10 illustrates LWD's training data infrastructure, which links a fleet of robot actors to a multi-host learner via a versioned-snapshot data plane. On the actor side, each robot runs an edge client that accumulates per-frame observations into complete episodes and uploads them to distributed object storage at episode boundaries; episode metadata is persisted by a business service and event notifications are published to a message queue.

On the cloud side, a central *Coordinator* consumes event notifications from the message queue, fetches episode metadata from object storage, and commits monotonically increasing snapshot versions that define the training data view at each step. The learner runs as a multi-host SPMD JAX program, with one process per node driving all local accelerators. Each process instantiates a *Distributed Replay Buffer (DRB) Reader* as its dataset; before each training step, all DRB Readers synchronize on the same snapshot version via a cross-host barrier, ensuring the SPMD collective sees a globally consistent dataset view despite asynchronous edge ingestion. Each DRB Reader spawns a prefetcher subprocess that downloads payloads from object storage in parallel; placing one prefetcher per node is sufficient to saturate the per-node read bandwidth available from the underlying distributed filesystem in our deployment.

Model parameters produced by the SPMD collective are published to a publish-subscribe channel that fans out to all robot actors, which reload the new policy at episode boundaries. Across the entire design, the Coordinator is the only orchestration singleton; both the actor fleet and the learner scale independently.

We characterize this infrastructure along two operational axes that are critical for online RL: whether every collected episode is reliably incorporated into training, and how quickly new data and updated policies traverse the actor–learner loop.

*1) End-to-End Reliability:* The system provides at-least-once end-to-end delivery for every episode produced on the actor side. (i) Object-storage uploads commit atomically (readers see either the fully-uploaded payload or no object) and are retried until persisted. (ii) Episode metadata is committed via a transactional insert in the business service, then announced to a durable message queue with delivery acknowledgment, so notifications survive coordinator restarts. (iii) Per-node prefetcher download tasks are requeued on failure with bounded retries; on snapshot commit, the snapshot data and the version pointer are updated atomically, so partial failures cannot leave a snapshot inconsistent. In our profiled 8-hour, 16-actor run of 1,604 episodes, every episode ingested in steady state completed the full end-to-end pipeline.

TABLE IV

**ABLATION OF VALUE LEARNING DESIGN (COMPLETE RESULTS).** COMPLETE RESULTS ON GROCERY RESTOCKING TASKS AND LONG-HORIZON TASKS. WE COMPARE SCALAR EXPECTILE REGRESSION AND OUR DISTRIBUTIONAL IMPLICIT VALUE LEARNING UNDER OFFLINE AND ONLINE SETTINGS. WE REPORT TASK SUCCESS RATE FOR EACH TASK AND THE AVERAGE ACROSS ALL EIGHT TASKS. THE BEST RESULT PER COLUMN IS SHOWN IN BOLD.

| Method | | Grocery Restocking Tasks | | | | Long-Horizon Tasks | | | | Average |
|---|---|---|---|---|---|---|---|---|---|---|
| | | Restocking | Correction | Freezer | Open-Cooler | Gongfu Tea | Fruit Juice | Cocktail | Shoebox | |
| Offline | Expectile Regression | 0.85 | **1.00** | **1.00** | **1.00** | 0.68 | 0.74 | 0.71 | 0.76 | 0.84 |
| | DIVL (ours) | **1.00** | **1.00** | 0.92 | 0.95 | 0.72 | 0.74 | 0.83 | 0.86 | 0.88 |
| Online | Expectile Regression | 0.95 | **1.00** | 0.92 | **1.00** | 0.76 | 0.76 | 0.77 | 0.84 | 0.88 |
| | **DIVL (ours)** | **1.00** | **1.00** | 0.97 | 0.98 | **0.89** | **0.90** | **0.93** | **0.92** | **0.95** |

TABLE V

**ABLATION OF THE ADAPTIVE $\tau$ STRATEGY IN OFFLINE LWD.** WE COMPARE THE ADAPTIVE $\tau$ SCHEDULE WITH A CONSTANT $\tau$ BASELINE. FOR THE CONSTANT BASELINE, $\tau$ IS SET TO THE EMPIRICAL AVERAGE VALUE OF THE ADAPTIVE SCHEDULE FROM THE ADAPTIVE-$\tau$ RUN ($\tau = 0.52$), WHILE ALL OTHER TRAINING COMPONENTS ARE KEPT UNCHANGED.

| Method | Grocery Restocking Tasks | | | | Long-Horizon Tasks | | | | Average |
|---|---|---|---|---|---|---|---|---|---|
| | Restocking | Correction | Freezer | Open-Cooler | Gongfu Tea | Fruit Juice | Cocktail | Shoebox | |
| LWD Offline, constant $\tau$ | 0.85 | 0.88 | **0.94** | **0.95** | 0.70 | **0.76** | 0.70 | **0.90** | 0.84 |
| LWD Offline, adaptive $\tau$ | **1.00** | **1.00** | 0.92 | **0.95** | **0.72** | 0.74 | **0.83** | 0.86 | **0.88** |

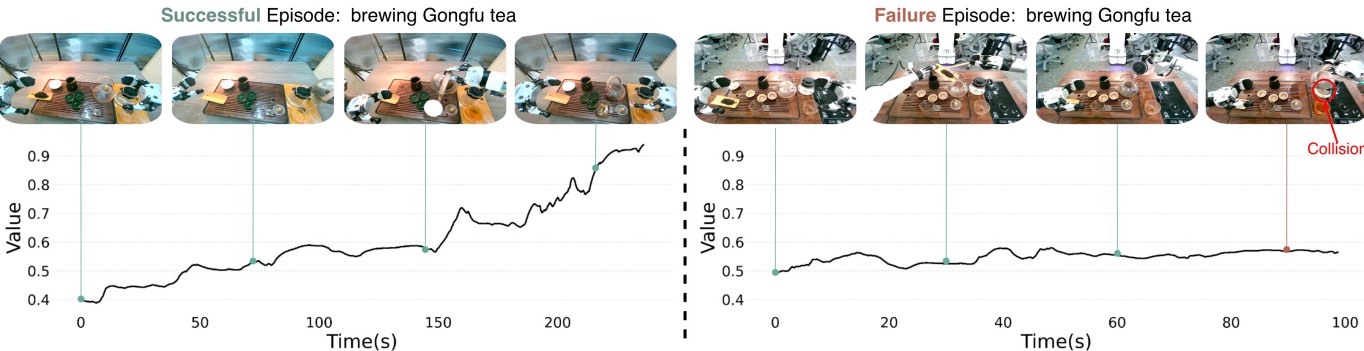

Fig. 8. **Value-learning visualization.** Quantile values of the learned distributional value function $V$ over representative Gongfu Tea episodes. The left trajectory succeeds and the right trajectory fails.

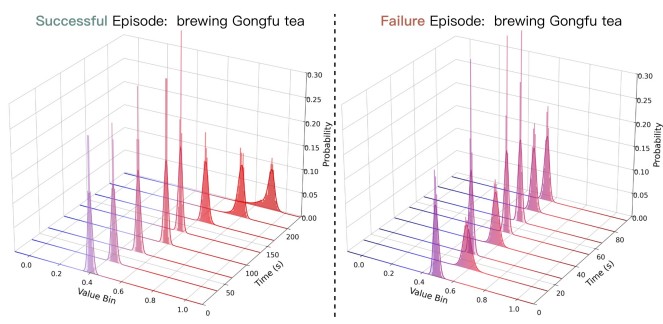

Fig. 9. **Predicted value distributions.** In the successful episode, the predicted distribution remains unimodal and its mode increases steadily from approximately 0.4 to 1.0. In contrast, the failure episode shows limited mode progression, rising only from approximately 0.5 to 0.6 before plateauing.

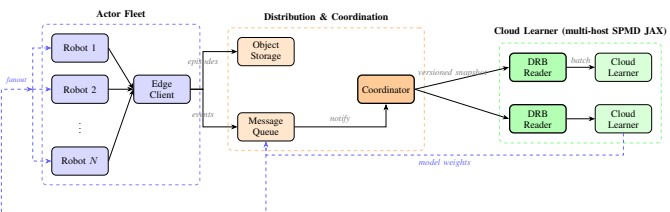

Fig. 10. **Distributed data infrastructure for LWD.** Robot actors upload episodes to object storage and publish event notifications to a message queue. A central *Coordinator* consumes notifications, fetches episode metadata, and commits versioned snapshots. The learner runs as a multi-host SPMD JAX program; on each node, the dataset (*DRB Reader*) holds a snapshot-bound view, spawns a prefetcher subprocess to download payloads from object storage, and feeds mini-batches to the local learner process. All DRB Readers synchronize on the same snapshot via a cross-host barrier. Updated model parameters produced by the collective are published back to all robot actors via the message-queue-backed publish-subscribe channel.

*2) Operational Latency:* We report the two end-to-end latencies that govern the tightness of the actor-learner loop: (i) *episode-to-learner*: the elapsed time from when an episode is produced on an actor to when it becomes available for the learner to sample; and (ii) *model-to-actor*: the elapsed time from when the learner publishes a new policy to when the

TABLE VI
**OPERATIONAL LATENCY.** END-TO-END LATENCY MEASURED ON THE
SAME 8-HOUR, 16-ACTOR ONLINE-RL RUN AS THE END-TO-END
RELIABILITY SUBSECTION ABOVE. ABSOLUTE VALUES ARE SENSITIVE TO
NETWORK CONFIGURATION AND LINK CONTENTION AND MAY VARY
ACROSS DEPLOYMENTS.

| Path | P50 | P99 |
|------|-----|-----|
| Episode produced $\rightarrow$ available to learner | 41 s | 148 s |
| Model published $\rightarrow$ received by actor | 38 s | 55 s |

actor has loaded it for the next rollout. Table VI reports both
on the same 8-hour, 16-actor run as the End-to-End Reliability
subsection above. Both latencies are dominated by object-
storage I/O on the actor-to-cloud link — the episode payload
in one direction, the policy artifact in the other — so absolute
values are sensitive to link bandwidth and contention and may
vary substantially across deployments.