# OpenReview forum: "Learning While Deploying: Fleet-Scale Reinforcement Learning for Generalist Robot Policies"
_roboticsfoundation.org/RSS/2026/Workshop/RL4VLA — RL4VLA_

### Official Review · Reviewer_3pdd · 2026-06-28
**Practical and timely paper with impressive real-world results**

**Rating:** 7
**Confidence:** 3

**Review:**

- The paper proposes Learning While Deploying (LWD), a fleet-scale offline-to-online RL framework. LWD includes successes, failures, and interventions. Since fleet data is often multi-modal, they propose Distributional Implicit Value Learning (DIVL), which is a distributional value model that builds on value-learning of Implicit Q-Learning. They combine it with QAM, or Q-learning via adjoint matching, to update a flow-based VLA policy. They try to improve the generalist policy across a wide range of tasks simultaneously.
- Pros:
    - Practical real-world system with strong baselines and experiments (eight real-world tasks that are short and long-horizon)
    - Distributional value model seems to have benefits for fleet learning and is well-motivated. They have an adaptive optimistic mechanism such that for more confident value distributions, they are more optimistic.
- Cons:
    - Should report full-task (binary) success for long-horizon tasks (not just step-wise success). The overall average success rate also combines binary success rates with step-wise scores. There could also be more statistical details (such as number of episodes / trials per task, confidence intervals, etc.).
    - Since updated policies are periodically redeployed to the fleet, the paper could discuss the safety of this.
    - Could have more discussion if there is any issues with catastrophic forgetting across tasks during online updates.
- Overall, this is a strong paper with high potential for real-world deployment and use.

---

### Official Review · Reviewer_Y4Q4 · 2026-06-29
**Review on Learning While Deploying**

**Rating:** 8
**Confidence:** 4

**Review:**

The paper introduces LWD, a fleet-scale offline-to-online RL method for post-training generalist VLA policies from deployment rollouts and human interventions. On 16 dual-arm robots and eight real-world tasks, a single policy improves with fleet data to 95% average success, with the largest gains on long-horizon tasks.

Good motivation and clear presentation. Extensive Real World Experiments with good results on challenging long-horizon tasks. I would only have minor negative comments. The paper exceeds my expectation for a workshop.

---

### Decision · Program_Chairs · 2026-07-03

**Decision:**

Accept

**Comment:**

This paper presents a rare, well-executed real-world study of offline-to-online reinforcement learning for post-training a generalist vision-language-action model, demonstrating improvements across 16 dual-arm robots and 8 real-world tasks, with particularly strong gains on long-horizon manipulation. The reviewers agreed that the scale and practical significance of the work are major strengths and both recommended acceptance. The main concerns are limited statistical reporting and the use of mixed binary and step-wise evaluation metrics. We believe these issues do not outweigh the overall contribution, and the paper represents a valuable addition to the workshop. For the camera-ready version, the authors should report full-task binary success separately from step-wise scores for long-horizon tasks, include the already-collected episode counts and confidence intervals, use a consistent method name throughout the paper (LWD vs. Fleet-Scale), and briefly discuss redeployment safety and potential forgetting.